# Soil Fungal Composition Drives Ecosystem Multifunctionality after Long-Term Field Nitrogen and Phosphorus Addition in Alpine Meadows on the Tibetan Plateau

**DOI:** 10.3390/plants11212893

**Published:** 2022-10-28

**Authors:** Bingheng Cheng, Hongyan Liu, Juan Bai, Jinhua Li

**Affiliations:** State Key Laboratory of Herbage Improvement and Grassland Agro-Ecosystems, College of Ecology, Lanzhou University, Lanzhou 730000, China

**Keywords:** nutrient addition, species richness, community composition, abiotic factors, ecosystem functioning

## Abstract

An ecosystem can provide multiple functions and services at the same time, i.e., ecosystem multifunctionality (EMF). Above- and belowground biodiversity and abiotic factors have different effects on EMF. Human activities increase atmospheric nitrogen (N) and phosphorus (P) deposition, but the mechanism of how atmospheric N and P deposition affect EMF in alpine meadows on the Tibetan Plateau is still unclear. Here, we measured eleven ecosystem parameters to quantify EMF by averaging method and explored the impact of plant and microbial species diversity and abiotic factors on EMF after long-term field N and P addition in alpine meadows on the Tibetan Plateau. Results showed that N addition reduced EMF by 15%, NP increased EMF by 20%, and there was no change due to P addition. N and P addition reduced pH, relative light conditions (RLC), and plant species richness and modified plant and fungal community composition. Structural equation model (SEM) analysis confirmed that fungal community composition was an important and positive driver on EMF. These results provided an understanding of how N and P addition affect EMF directly and indirectly through biotic and abiotic pathways, which was important for predicting the response of EMF to atmospheric N and P deposition in the future. Furthermore, the findings suggested that soil fungal composition was more important driving factors than abiotic factors in the response of EMF to N and P addition and the importance of the interactions between plant and soil microbial species diversity in supporting greater EMF.

## 1. Introduction

Ecosystem can provide multiple functions and services at the same time, that is, ecosystem multifunctionality (EMF), which is usually calculated by the averaging method, single threshold method or multiple-threshold method [1]. The relationship between biodiversity and ecosystem multifunctionality (BEMF) has become a new research hotspot in recent years. Species diversity is widely used in BEMF [1,2,3,4,5,6]. For example, Hector and Bagchi [7] quantitatively analyzed the impact of species diversity on EMF and concluded that maintaining higher EMF requires more species. Maestre et al. [8,9] indicated that species diversity was an important driving factor for the EMF of dryland ecosystems. Different scales of species diversity have different effects on EMF. Based on the results of long-term landscape experiments, α diversity had stronger positive effects on most of the individual functions and EMF than β and γ diversity, and higher β diversity was not conducive to the variability of EMF [10]. α diversity and β diversity are important driving factors of EMF at larger spatial scales in European forest landscapes [11]. In ecosystems structured largely by natural processes, β diversity was a more important driver than α diversity in improving EMF and highlighted the importance of community heterogeneity in the process of ecosystem restoration [12].

Soil microorganisms play a key role in maintaining EMF, as most soil functions are microbially mediated [13,14,15]. Fungi and bacteria, as two major groups of microorganisms, have different effects on EMF. Soil bacterial diversity was much higher than fungal diversity and was positively correlated with EMF in alpine ecosystems [13]. Mycorrhizal fungi were major drivers of nutrient cycles in terrestrial ecosystems and played a key role in maintaining higher EMF [16]. In dryland, fungi were more tolerant to drought than bacteria and hence contributed more to EMF [17]. Wagg et al. [14] linked microbial co-existence networks with EMF, revealed the importance of interactions within fungi and bacteria to improve EMF, and proved that the disappearance of microbial interactions would damage ecosystem functions. In the boreal forest, complex carbon (C) is mainly decomposed by fungi and fungal diversity has the highest impact on EMF [18].

Increasing studies have shown that the interactions between aboveground and underground communities play an important role in improving ecosystem functions and adapting to environmental changes [19,20]. Under extreme drought scenarios, positive associations between plants and microbes can enhance tolerance to abiotic stress [21]. A review indicated that the complex interactions between above- and belowground communities regulate the impact of plant diversity on productivity and soil nutrients [22]. The associations between plants and arbuscular mycorrhizal fungi largely facilitate the accumulation of soil C and nitrogen (N) contents in the plant rhizosphere [23]. Similarly, Cui et al. [24] indicated that the loss of above- and belowground biodiversity enhanced the direct negative impact of grassland degradation on EMF and highlighted the importance of their association in supporting greater grassland EMF.

Alpine meadow is the most widely distributed grassland type on the Qinghai Tibet Plateau. It plays an important role in maintaining higher biodiversity and EMF, improving ecosystem stability, and promoting economic development. However, human activities have increased atmospheric N and phosphorus (P) deposition, and high N deposition areas in China are gradually spreading from southeast to northwest [25], which will have an unpredictable impact on biodiversity and EMF in the future. At present, the mechanism by which atmospheric N and P deposition affect EMF in alpine meadows on the Tibetan Plateau is still unclear, which will limit our ability to accurately predict the change in EMF under atmospheric N and P deposition.

In this study, we explored the impact of plant and microbial species diversity and abiotic factors on EMF after long-term field N and P additions to alpine meadows. Our study aimed at answering the following questions: (1) How does N and P addition affect the EMF in alpine meadows on the Tibetan Plateau? (2) The relative importance of plant and microbial species diversity and abiotic factors on the EMF. The results will provide reference for predicting the response of EMF to atmospheric N and P deposition in the future.

## 2. Results

### 2.1. Effects of N and P Addition on Ecosystem Parameters

Compared to the Control, N addition reduced EMF by 15%, NP significantly increased EMF by 20%, and there was no change due to P addition (Figure 1l). Specifically, N addition significantly increased aboveground biomass (AGB), belowground biomass (BGB), soil organic carbon (SOC), NO_3_^−^−N, and roots total nitrogen (RTN) by 110%, 46%, 12%, 33%, and 24%, respectively, but reduced microbial biomass carbon (MBC), cumulative carbon mineralization (CCM), and soil available phosphorus (SAP) by 46%, 37%, and 64%, respectively. P addition significantly increased microbial biomass nitrogen (MBN), RTN, soil ammonia nitrogen (NH_4_^+^−N), and SAP by 23%, 24%, 14%, and 92%, respectively, but reduced root organic carbon (ROC), MBC, CCM, and soil nitrate nitrogen (NO_3_^−^−N) by 12%, 15%, 29%, and 27%, respectively. NP addition significantly increased AGB, NO_3_^−^−N, RTN, and SAP by 120%, 30%, 33%, and 78%, respectively, but reduced NH_4_^+^−N by 18% (Figure 1a–k).

### 2.2. Effects of N and P Addition on Abiotic Factors and Plant and Microbial Species Diversity

Compared with the Control, N and P addition reduced pH by 0.4–0.5 unit (Figure 2a), and relative light conditions (RLC) were 13%, 25%, and 11% (Figure 2c) under N, P, and NP, respectively. N and P addition had no significant effects on soil bulk density (SBD) and soil water content (SWC) (Figure 2b,d).

N and P addition significantly reduced plant species richness with the lowest richness under NP addition (Figure 3a) and modified plant community composition (N and Control, *p* < 0.05; P and Control, *p* < 0.05; NP and Control, *p* < 0.01) (Figure 3b).

N and P addition had no significant effect on fungal and bacterial species richness (Figure 4a,b). However, N and P addition significantly modified fungal community composition (N and Control, *p* < 0.01; P and Control, *p* < 0.01; NP and Control, *p* < 0.01) (Figure 4c), and N addition significantly modified bacterial community composition (N and Control, *p* < 0.01) (Figure 4d).

### 2.3. Relationships between Abiotic Properties, Plant and Microbial Species Diversity, and Ecosystem Parameters

AGB and RTN were negatively correlated with pH and RLC, but MBC and CCM were positively correlated with pH and RLC. BGB and SOC were negatively correlated with RLC. EMF had no significant correlations with pH and RLC (Appendix A).

Plant species richness and plant and fungal composition were positively correlated with RLC, and plant species richness was also positively correlated with pH (Appendix A). Ecosystem parameters (except MBN) and species richness had no significant correlations with SBD and SWC (Appendix A).

AGB, NO_3_^−^−N and RTN were negatively correlated with plant species richness, while NH_4_^+^−N was positively correlated with plant species richness. AGB, MBC, and SAP were positively correlated with plant composition (Figure 5, Appendix A).

Ecosystem parameters had no significant correlations with microbial species richness (Appendix A). BGB, MBC, SOC, MBN, NO_3_^−^−N, and SAP were positively correlated with bacterial composition. AGB, ROC, MBC, MBN, NO_3_^−^−N, NH_4_^+^−N, and SAP were positively correlated with fungal composition. EMF had no significant correlations with microbial composition (Figure 5, Appendix A).

### 2.4. The Direct and Indirect Mediation Effect of N and P Addition on EMF

Our structural equation model (SEM) explained 75% of the variation in the EMF, with no significant direct effect of N and P addition on EMF. The model was satisfactorily fitted to our data (Figure 6a). N and P addition had negative effects on pH, RLC, and plant species richness (except P addition) while having positive effects on fungal and plant composition (except N addition). RLC had negative effects on plant composition and pH had positive effects on fungal composition. Plant species richness had negative effects on fungal composition. Fungal composition had positive effects on EMF (Figure 6a). NP addition showed positive effects on EMF while N or P addition showed negative effects. Plant and microbial species diversity were more important driving factors than abiotic factors in the response of EMF to N and P addition. Fungal composition was the most important and positive factor in fostering EMF, followed by pH, RLC, and plant composition (Figure 6b).

## 3. Discussion

Our study indicated that NP increased EMF, while N reduced EMF, and P had no effect on EMF. The impacts of N and P addition on EMF were mediated by abiotic factors and plant and microbial species diversity in alpine meadows on the Tibetan Plateau. In addition, the fungal composition was the most positive factor in improving EMF.

### 3.1. The Responses of Abiotic Factors to N and P Addition

As many studies have shown the acidification by nutrient additions [6,26,27,28,29,30,31], our results showed that N and P addition reduced soil pH. The decline in pH could be explained by (1) plants growing rapidly and producing a large amount of organic acids, such as malic acid and citric acid; (2) soil microorganisms producing a large amount of H^+^, NO_3_^−^, and NH_4_^+^ through ammonification and nitrification while plants and soil microorganisms were unable to absorb enough H^+^, NO_3_^−^, and NH_4_^+^, resulting in the accumulation of these irons in the soil and a decrease in pH [32,33].

N and P addition reduced RLC levels in the lower canopy layers, with the largest reduction under NP addition. The decline in RLC could be due to N and P addition promoting plant growth, indicated by higher plant biomass, leaf area, stem height, and the average height and canopy density of the aboveground plant community, thus decreasing luminous flux transmitted to the lower canopy layers [34].

### 3.2. The Responses of Plant and Microbial Species Diversity to N and P Addition

In line with the findings on the effects of nutrient addition on plant communities [6,26,28,34], our results showed that N and P addition significantly reduced plant species richness and modified plant community composition (Figure 3). These changes in the plant community were mainly ascribed to light competition [34]. N and P addition alleviated the soil nutrient limitation and promoted plant growth, in particular for some forbs and grasses, which grew rapidly with developed root systems and high nutrient utilization efficiency, making the stems thicker and taller and the leaf area larger. These plants occupied most of the space in the community and increased the density and height of the community canopy, resulting in a significant reduction in the luminous flux at the lower canopy layers, which had a shading effect on dwarf plants. In addition, the dominant plants distributed photosynthetic products to the aboveground parts for rapid growth to compete for more light resources. While significantly improving the aboveground biomass, much litter covered the lower layer of the community, which not only affected the germination of plant seeds and the survival of seedlings, but also intensified the shading effect on dwarf plants [35], resulting in the loss of a large number of shade-intolerant plants. In our study site, N addition led to the loss of most legumes and a small number of forbs compared to the Control. The plant community was dominated by grasses, sedges, and some forbs. The NP addition led to the loss of all legumes, sedges, and most forbs. The plant community were dominated by grasses and tall forbs. In other words, grasses became the predominant species due to their competitive advantages in light and nutrient utilization.

Our results showed that N and P addition had no significant effect on microbial species richness but significantly modified fungal composition, indicating that fungal communities were more sensitive to N and P addition than bacterial communities, which was consistent with recent study results [36,37,38]. Increasing studies have shown that soil pH is one of the most important determinants of microbial community composition [37,39,40,41,42]. In addition, the changes in plant composition and soil available nutrients may be responsible for the sensitive responses of fungal community to N and P addition. Fungal and bacterial biomass have different C:N ratios, with C:N ratios around 5 for bacteria and around 10 for fungi in soil [43,44]. In general, fungi and bacteria prefer substrates with different C:N ratios, with fungi preferring complex C sources with high C:N ratios in comparison to bacteria [45,46]. In our study, soil dissolved organic C (DOC): dissolved organic nitrogen (DON) increased significantly (Appendix A), and N and P addition made the grasses with higher C:N ratios (Appendix A) become the dominant species, dominate the community in productivity, and thus input C into the soil through plant litter. These were beneficial to the growth of fungi, evidenced by the highly correlation between plant diversity and fungal composition (Figure 6a and Appendix A), as found in other grasslands [37,47,48].

### 3.3. The Responses of Ecosystem Parameters to N and P Addition

Due to the slow mineralization of soil organic matter, the content of soil available nutrients (available N, available P etc.) is low in the Qinghai-Tibet Plateau [49,50]. Moreover, AGB and BGB increased significantly after N addition, which further indicated that plant growth in our study site was N-limited. Plants take inorganic N as the main N source, and most plants in the alpine meadow on the Qinghai Tibet Plateau prefer to absorb NO_3_^−^-N [51,52,53]. In our study, N addition increased NO_3_^−^−N and alleviated the influence of N limitation on plant growth, which promoted plant growth and improved AGB, BGB and RTN. The decrease in soil pH reduced the activity of microbial enzymes, the stability of cell membranes, and the content of mineral elements, such as Ca^2+^ and K^+^ [54], thereby reducing MBC and CCM. However, the response of microbial growth to N addition cannot be explained only by changes in MBC, and microbial growth and soil respiration must be decoupled [55]. In addition, N input can modify the priming effect and accelerate mineralization of native SOM [56,57]. In our study, DOC, MBC, RTN, and NO_3_^−^−N increased while SAP and DOP decreased in N addition, suggesting that N addition can accelerate organic matter decomposition and nutrient release rate, and improve soil available nutrients and biological absorption of nutrients [58]. Meanwhile, the increase in C input and the decrease in C output increased the SOC. P addition increased SAP and stimulated organisms to absorb more N, which was reflected in the increase of MBN and RTN. P addition reduced ROC, MBC, CCM, and NO_3_^−^−N, which could be explained by the aggravation of N limitation and the decrease in soil pH. NP addition increased most of ecosystem parameters, indicating that NP can truly alleviate the impact of soil nutrient restriction on the growth of plants and microorganisms and the soil acidification caused by N and P addition [59].

### 3.4. The Effect of the Direct and Indirect Pathways of N and P Addition on EMF

Our SEM showed that the effects of N and P addition on EMF were mainly dominated by indirect pathways. Specifically, for the aboveground indirect pathway, N and P addition alleviated the limitation of soil nutrients, changed the competition pattern between plants (from underground nutrient competition to aboveground light resource competition) and plant community composition (reflected in that grasses became the dominant group and dominated resource utilization and dry matter accumulation) by reducing RLC and increased AGB, BGB, and EMF. For the underground indirect pathway, N and P addition modified fungal community composition by reducing soil pH and modified plant community composition and the quantity and quality of the litter. N and P addition improved litter decomposition, nutrient cycling speed, and available nutrient content, which increased the absorption of nutrients by plants and further increased the AGB, BGB, and EMF. Compared with abiotic factors, plant and microbial species diversity were more important driving factors in supporting greater EMF, which was consistent with previous research results [14,20,24,60,61]. In particular, aboveground biodiversity modified underground biodiversity by increasing the quantity and quality of litter and changing the total amount of soil nutrients [62]. On the contrary, underground biodiversity improved aboveground biodiversity and increased aboveground biomass by accelerating litter decomposition, improving nutrient cycling speed and available nutrient content [14].

## 4. Materials and Methods

### 4.1. Study Site and Experimental Design

The study site (N34°55′, E102°53′, 3000 m above sea level) is located at the Gannan Grassland Ecosystem Field Observation and Research Station on the Tibetan Plateau, Hezuo City, Gannan Tibetan Autonomous Prefecture, Gansu Province. The climate is the plateau continental monsoon climate, and the winter is long, dry, and windy. The annual average temperature is 2 °C. The average annual precipitation is 558 mm, concentrated in July, August, and September, and the annual evaporation is 1222 mm. The soil is classified as chestnut soil according to the FAO soil classification system or sub-alpine meadow soil according to the Chinese soil classification system [26,29]. The thickness of soil horizons ranges from 50 to 100 cm, and the soil texture is light loam. Soil pH ranges from 7 to 7.5. Soil organic C content, total nitrogen content, and total phosphorus content are 33.8 ± 1.8 g kg^−1^, 3.7 ± 0.1 g kg^−1^, and 0.65 ± 0.01 g kg^−1^, respectively, in 0–20 cm soil horizons [26]. The vegetation is a typical alpine meadow, dominated by grasses such as *Poa annua* Linn., *Elymus nutans* Griseb., and *Festuca ovina* Linn.; sedges such as *Scirpus pumilus* Vahl, *Kobresia capillifolia* (Decne.) C. B. Clarke, *Kobresia myosuroides* (Villars) Foiri., *Scirpus triqueter* Linn., and *Kobresia myosuroides* (Villars) Foiri.; forbs such as *Anemonerivularis* Buch.-Ham., *Taraxacum lugubre* Dahlst, *Geranium wilfordii* Maxim., *Artemisia polybotryoidea*., *Thalictrum aquilegiifolium* var. *sibiricum* Linnaeus, *Euphorbia esula* Linn., *Stellera chamaejasme* Linn., *Gentiana farreri* Balf. f.; and legumes such as *Oxytropis ochrocephala* Bunge, *Gueldenstaedtia verna* (Georgi) Boriss, *Medicago falcata* Linn., and *Vicia sepium* Linn.

The field plots were set up for four treatments: N-alone addition, P-alone addition, N and P together addition, and Control without any addition. Each treatment had five replicates (5 × 5 m each), a total of 20 plots and 10 g m^−2^ yr^−1^ of N and/or P in the form of urea and sodium dihydrogen phosphate anhydrous were added annually starting in 2009 [27].

### 4.2. Plant and Soil Sampling

In early August of 2020, we randomly selected 50 × 50 cm quadrat from the center of each plot to collect plant and soil samples during the peak plant growth season (July–August). We investigated the number of plant species and average height and coverage of each plant in each quadrat and then used scissors to harvest plant samples 1 cm above ground level. Samples were then sorted by species. After clipping the aboveground plant, we used a drill to collect randomly three soil cores (5 cm diameter by 20 cm depth) and then mixed them and washed them gently with water over a 60 mesh screen until roots were separated from the soil in each quadrat, thus obtaining root samples. All plant and root samples were dried at 65 °C for 48 h and weighed as AGB and BGB, respectively (g m^−2^). The dried root samples were left for measuring the ROC and RTN.

Before clipping the aboveground plant, we randomly selected ten sites in each plot (including quadrat) to measure the upper (0.5 m higher than the canopy) and lower (at 1 cm above the ground level) luminous flux values of the plant canopy with the illumination photometer (TES-1330A) on sunny days. RLC was obtained by calculating the ratio of the difference values (upper luminous flux minus lower luminous flux) to the upper luminous flux values.

We collected randomly five soil cores (5 cm diameter by 20 cm depth) in each plot (including quadrat) and then mixed them and sieved to 2 mm to remove roots. One portion was stored at 4 °C for the measurement of chemical properties; the other was kept at −80 °C for microbial sequencing.

### 4.3. Soil and Root Samples Properties Analysis and Microbial High-Throughput Sequencing

Soil pH, SBD, and SWC were measured using standard methods as described in Cui et al. [24]. ROC, RTN, SOC, MBC, MBN, CCM (cumulative C mineralization), NH_4_^+^−N, and NO_3_^−^−N were measured using standard methods as described in Li et al. [26]. Soil DOC and DON were extracted with 0.5 M K_2_SO_4_ in a ratio of 1:5 via shaking at 200 rpm for 1 h followed by filtration through 0.45 μm polytetrafluoroethylene filters [6,63,64]. SAP was measured with sodium hydrogen carbonate solution-Mo-Sb anti spectrophotometric method. The taxonomic compositions of bacterial and fungal communities were analyzed based on high-throughput sequencing of the 16S rRNA gene for the bacteria and the internal transcribed spacer (ITS) regions for fungi using an Illumina Hiseq 2500 platform (Novogene, Beijing, China). The detailed procedure was as described in Chen et al. [31].

### 4.4. Quantifying Ecosystem Multifunctionality and Biodiversity Indices

Based on element cycling, nutrient supply and productivity, we calculated EMF with eleven common ecosystem parameters (no multicollinearity) via the averaging method, including AGB, BGB, ROC, RTN, SOC, MBC, NH_4_^+^−N, NO_3_^−^−N, MBN, SAP, and CCM, where AGB and BGB represented productivity and SAP, NH_4_^+^−N, NO_3_^−^−N, MBN, and RTN represented nutrient supply, ROC, SOC, MBC, and CCM represented C element cycling. These parameters represented different aspects of ecosystem functions and are often used in EMF research [17,65,66,67]. The averaging method quantified EMF by calculating the mean value of all ecosystem parameters with min–max normalization, using the following formulas:(1)$$\;EMF=\frac{\sum_{i=1}^{n}EP_{i}}{n}$$
(2)$$\;EP_{i}=\frac{X_{i}-min}{max-min}$$
where *EP_i_* is the standardized value of ecosystem parameter *i*, *n* is the number of parameters, *X_i_* is the ecosystem parameter *i* original sample value, *max* is the maximum value of all ecosystem parameter *i* sample values, and *min* is the minimum value of all ecosystem parameter *i* sample values.

Plant species richness was calculated as the total number of plants species in the quadrat. Principal component analysis (PCA) and analysis of similarities (Anosim) were carried out to qualitatively compare plant community composition between N and/or P addition. The variations of plant composition were quantified by PCA1 for SEM. Microbial species richness was calculated as the total number of operational taxonomic units (OTUs). Non-metric multidimensional scaling (NMDS) (using Bray–Curtis dissimilarity) and Anosim were used to qualitatively compare microbial community composition between N and/or P addition, and microbial composition variations were quantified using a Bray–Curtis dissimilarity matrix for SEM.

### 4.5. Statistical Analysis

A one-way analysis of variance (ANOVA) was used to test to assess the effects of N and P addition on the abiotic factors, ecosystem parameters, EMF, plant species richness and microbial species richness. Linear correlations were carried out to assess how abiotic factors correlated with ecosystem parameters, EMF, and species richness, and how species richness correlated with ecosystem parameters and EMF. The Mantel test was used to assess how plant or microbial communities correlated with ecosystem parameters, EMF, and abiotic factors. SEM was applied to evaluate the links between pH, RLC, plant and microbial biodiversity, and ecosystem multifunctionality after accounting for the attribution of multiple key ecosystem factors simultaneously [17] and to understand the causal pathways through which N and P addition influences EMF. Before modelling, we first employed a conceptual model (Appendix A) to demonstrate the relationship among different factors. Moreover, we calculated the total standardized effect of each driver on EMF. SEM was conducted by using AMOS 26 (IBM SPSS, Chicago, IL, USA). One-way ANOVA and linear correlations were conducted using IBM SPSS 26. PCA, NMDS, Anosim, and the Mantel test were conducted using R 3.6.1.

## 5. Conclusions

In summary, our results provided an understanding of how N and P addition affected ecosystem multifunctionality directly and indirectly through biotic and abiotic pathways, which was vital to predicting the response of EMF to atmospheric N and P deposition in the future. Furthermore, the findings suggested that plant and microbial species diversity, especially their interactions and soil fungal composition, were more important driving factors than abiotic factors in fostering EMF.

## Figures and Tables

**Figure 1 plants-11-02893-f001:**
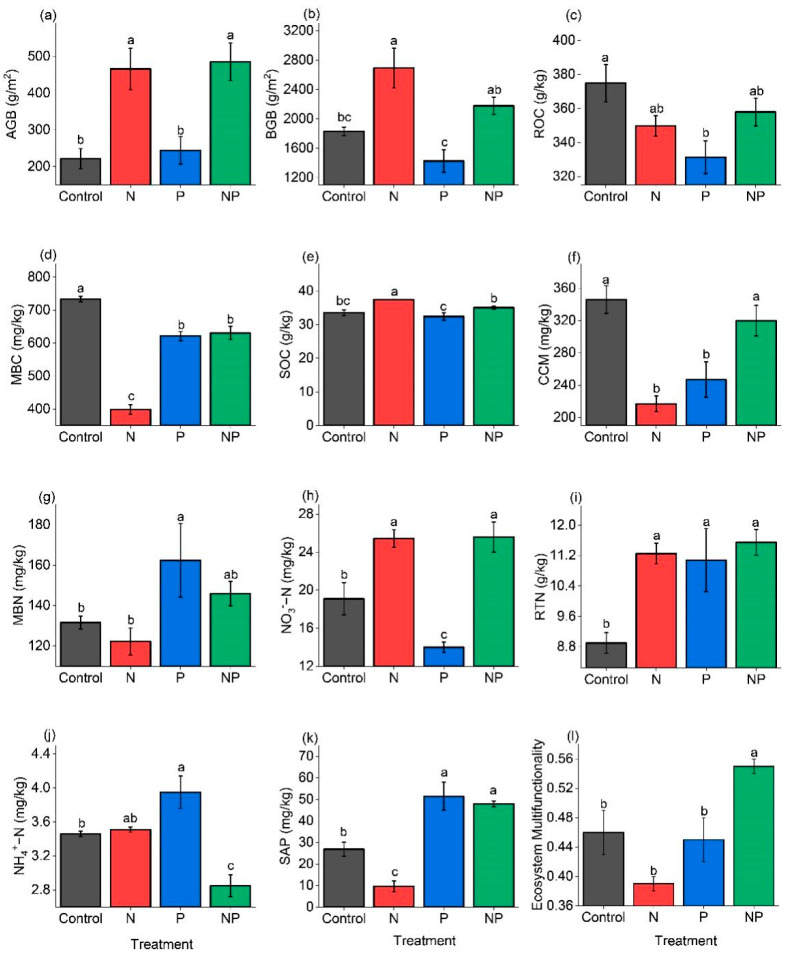
Effects of nitrogen and phosphorus addition on ecosystem parameters (M ± SE, *n* = 5), including (**a**) aboveground biomass (AGB); (**b**) belowground biomass (BGB); (**c**) roots organic carbon (ROC); (**d**) microbial biomass carbon (MBC); (**e**) soil organic carbon (SOC); (**f**) cumulative carbon mineralization (CCM); (**g**) microbial biomass nitrogen (MBN); (**h**) soil nitrate nitrogen (NO_3_^−^−N); (**i**) roots total nitrogen (RTN); (**j**) soil ammonia nitrogen (NH_4_^+^−N); (**k**) soil available phosphorus (SAP); (**l**) ecosystem multifunctionality (EMF). Different letters indicate that the same index has significant differences between different treatments (*p* < 0.05). Abbreviations: Control, the control without any nutrient addition; N, N-alone addition; P, P-alone addition; NP, N and P together addition.

**Figure 2 plants-11-02893-f002:**
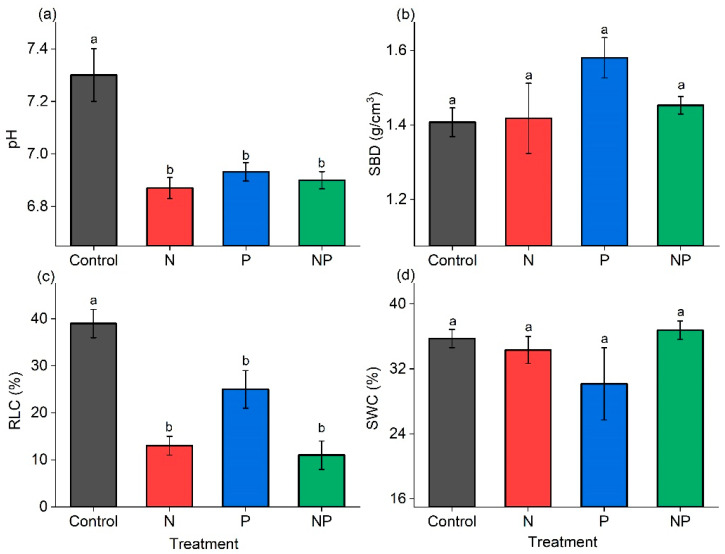
Effects of nitrogen and phosphorus addition on abiotic factors (M ± SE, *n* = 5), including (**a**) soil acidity and alkalinity (pH); (**b**) soil bulk density (SBD); (**c**) relative light conditions (RLC); (**d**) soil water content (SWC). Different letters indicate that the same index has significant differences between different treatments (*p* < 0.05). Abbreviations: Control, the control without any nutrient addition; N, N-alone addition; P, P-alone addition; NP, N and P together addition.

**Figure 3 plants-11-02893-f003:**
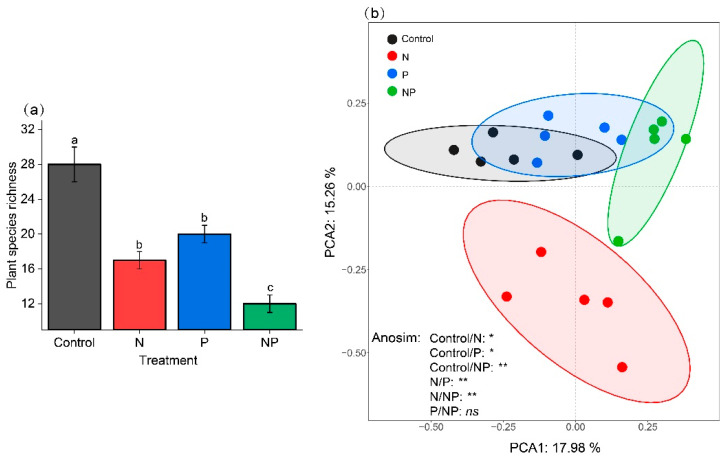
Effects of nitrogen and phosphorus addition on plant species richness (M ± SE, *n* = 5) and community composition, including (**a**) plant species richness; (**b**) plant community composition based on principal component analysis (PCA); and analysis of similarities (Anosim). Different letters indicate that the same index has significant differences between different treatments (*p* < 0.05). *, **, and ns indicated *p* < 0.05, *p* < 0.01, and *p* > 0.05, respectively. Abbreviations: Control, the control without any nutrient addition; N, N-alone addition; P, P-alone addition; NP, N and P together addition.

**Figure 4 plants-11-02893-f004:**
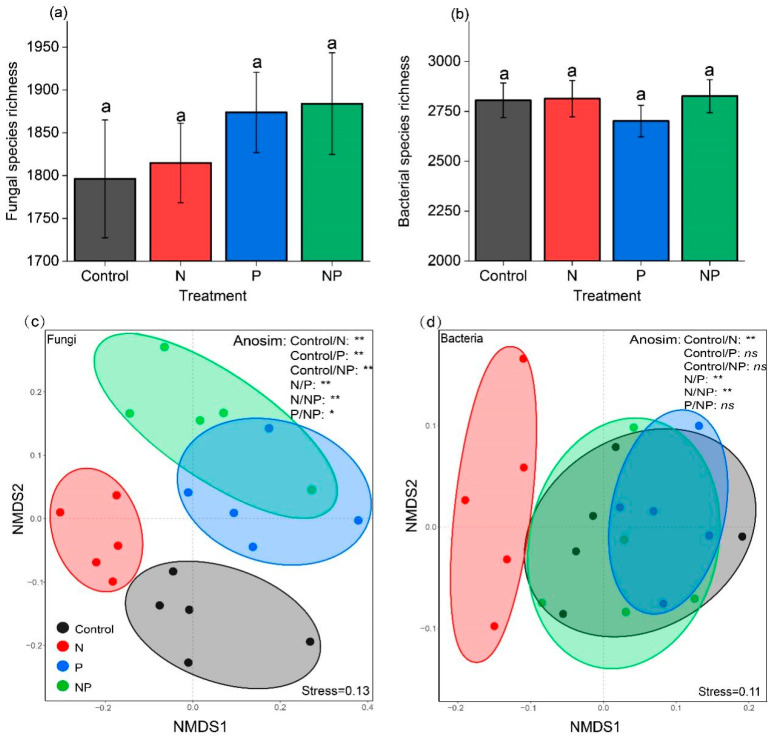
Effects of nitrogen and phosphorus addition on microbial species richness (M ± SE, *n* = 5) and community composition on operational taxonomic units (OTUs), including (**a**) fungal species richness; (**b**) bacterial species richness; (**c**) fungal community composition based on non-metric multidimensional scaling (NMDS) and analysis of similarities (Anosim) (**d**) bacterial community composition based on NMDS and Anosim. Different letters indicate that the same index has significant differences among different treatments (*p* < 0.05). *, **, and ns indicated *p* < 0.05, *p* < 0.01, and *p* > 0.05, respectively. Abbreviations: Control, the control without any nutrient addition; N, N-alone addition; P, P-alone addition; NP, N and P together addition.

**Figure 5 plants-11-02893-f005:**
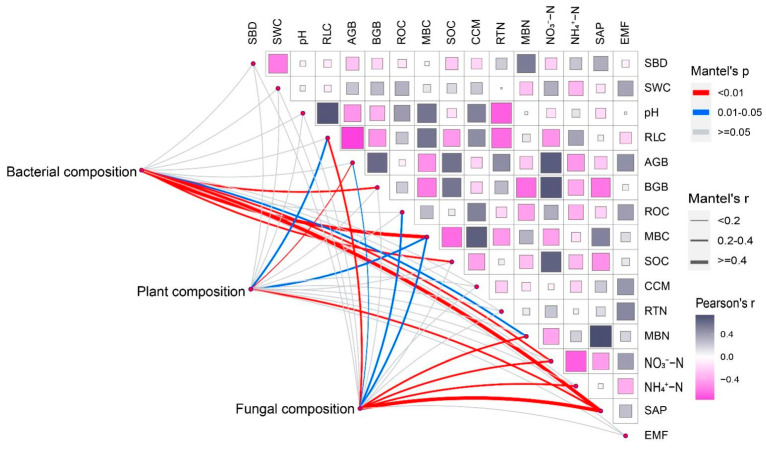
Mantel test analysis of ecosystem parameters and abiotic factors and plant and microbial composition on operational taxonomic units (OTUs). Abbreviations: pH, soil acidity and alkalinity; SBD, soil bulk density; SWC, soil water content; RLC, relative light conditions; AGB, aboveground biomass; BGB, belowground biomass; ROC, roots organic carbon; MBC, microbial biomass carbon; SOC, soil organic carbon; CCM, cumulative carbon mineralization; RTN, roots total nitrogen; MBN, microbial biomass nitrogen; NO_3_^−^−N, soil nitrate nitrogen; NH_4_^+^−N, soil ammonia nitrogen; SAP, soil available phosphorus; EMF, ecosystem multifunctionality.

**Figure 6 plants-11-02893-f006:**
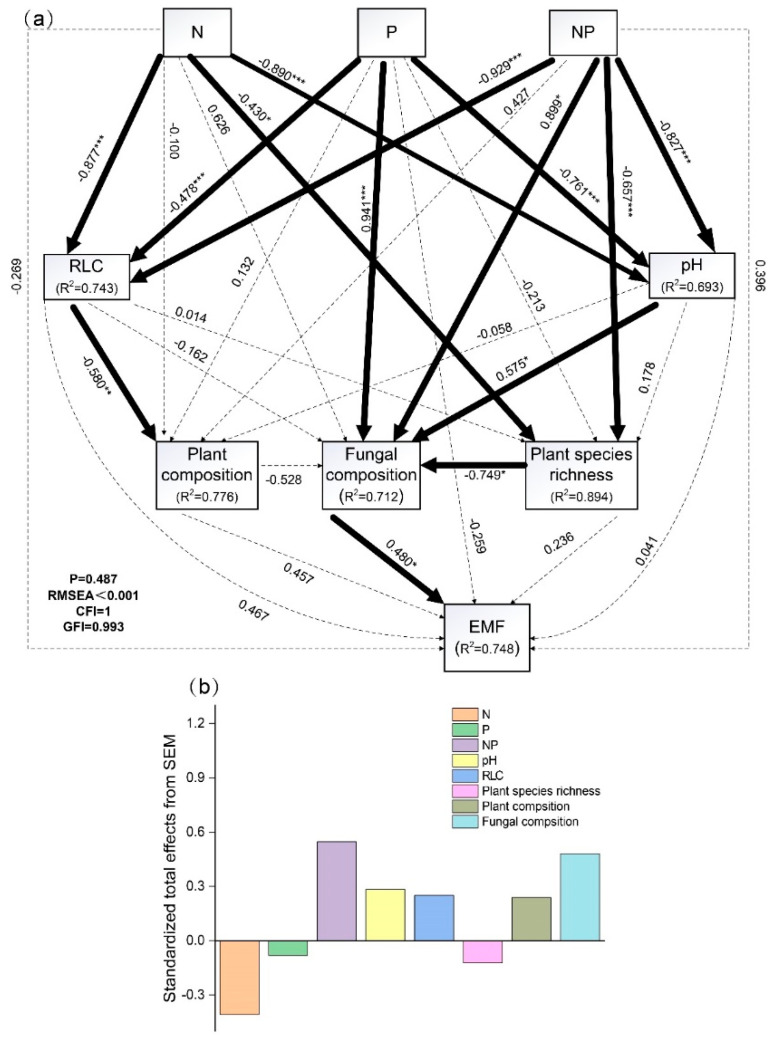
Structural equation model (SEM) of the effect of nitrogen and phosphorus addition on ecosystem multifunctionality, including (**a**) structural equation model; (**b**) standardized total effects from SEM. The solid line represents the path with significant influence (*, **, and *** indicated *p* < 0.05, *p* < 0.01, and *p* < 0.001, respectively), the dotted line represents the path with no significant influence, the number on the solid and dotted line represents the standardized path coefficient, and the R^2^ value represents the proportion of variation explained by causal relationships with other variables. Abbreviations: Control, the control without any nutrient addition; N, N-alone addition; P, P-alone addition; NP, N and P together addition; pH, soil acidity and alkalinity; RLC, relative light conditions; EMF, ecosystem multifunctionality; P, probability level; RMSEA, root mean square error of approximation; CFI, Comparative fit index; GFI, goodness-of-fit index.

## Data Availability

Not applicable.

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
