# Peer review of "Soil Fungal Composition Drives Ecosystem Multifunctionality after Long-Term Field Nitrogen and Phosphorus Addition in Alpine Meadows on the Tibetan Plateau"

_plants, 2022, doi:10.3390/plants11212893_

Round 1
Reviewer 1 Report
Keywords: Please avoid using words already mentioned in the title eg nitrogen and phosphorus addition; above- and belowground biodiversity; ecosystem multifunctionality
line 37: most of the individual functions
line 38: was not conducive to the variability
lines 40-41: β diversity was a more important
line 46: in alpine ecosystems
line 52: In the boreal forest,
lines 55-56: the aboveground and underground communities
line 78: How does N and P addition affect the....\
line 143: Plant species richness and plant and fungal composition
line 145: and species richness
line 165: with no significant direct effect
line 169 and elsewhere in the same paragraph: PAR had negative effects on plant composition and pH had positive effects on the fungal ...
line 191: the fungal ....
line 193: have shown the acidification
line 203: thus decreasing light radiation
line 206: In line with the findings on the effects
lines 208-209: These changes in the plant community were mainly ascribed to light competition
line 218: much litter covered the
lines 223 to 227: Pleasealways cite the species authors when it is first mentioned in your text
line 223: why Kobresiaceae and Scirpoideae? Was it impossible to identify the genus name?
lines 230-231: indicating that fungal communities were more sensitive to N and P addition than bacterial communities
line 246: Due to the slow mineralization of soil organic matter,
line 264: by the aggravation of N limitation
line 278: composition and the quantity and quality of the litter.
line 296: Field plots included N and P additions in the form of urea and sodium dihydrogen
line 311: Soils from the surface layer (0–20 cm depth) were collected randomly using an auger
line 345: were carried out to compare plant communities
line 347: to compare microbial communities
lines 355-356: In our SEM, plant communities were represented by PC1 and microbial communities
Author Response
Please see the attachment。

Reviewer 2 Report
The article has comprehensive statistically processed results of new rich experimental data from poorly known region of the Tibetan Plateau. It is a good research in plant ecology. This material shall be published. However the text should be corrected and improved.
1. A clear definition of the ecosystem multifunctionality shall be added in the Introduction with a short explanation on how it can be calculated. Figure 1L has numerical values of EMF, so the section 4.4 in the Material and Methods (M&M) must have functions/equations for EMF calculation.
2. The text contains numerous references to biodiversity, however M&M has no section on how biodiversity was quantitatively expressed, and the results on biodiversity evaluation are not visible in the text. Those seem strange, because this aspect has been under the scrutiny of researchers for a quite long time. Also strange is “belowground biodiversity” that relates only to roots, bacteria and fungi without consideration their taxonomic and functional characteristics. It is a main drawback of the article. Soil fauna as an important component of soil biodiversity is also missed in the article.
3. I was shocked when I saw Figure 2c before reading M&M. I understood from the figure that soil N and P can change solar radiation (PAR,%) going from our Sun! The authors think that they represented data on photosynthetically active radiation (PAR). It is wrong. Two lines in M&M (Section 4.1 L.303-304) show that it is a percentage of light reaching soil surface under plants’ canopy (cover), i.e. it is a measure of shadowing or plants’ density. It is not the abiotic factor; it is a parameter that related to plant community structure. The “PAR” shall be obligatory renamed. The text should be thoroughly corrected with other symbol (f.e. RPAR as relative PAR or RLC relative light conditions). Though, Discussion has a right explanation of the “PAR” effect on plant community (L 200-204).
Some specific comments
The text and supplementary data have many abbreviations for various parameters. I propose giving transcript of the abbreviations in a separate table.
L 18. SEM: please give a transcript of the abbreviation
L 83 “Effects of N and P addition on ecosystem functions”. Actually all numerical data below are parameters (state variables) but not functions (rate variables, flows)
L 86. If no special table for abbreviations, please give transcripts of abbreviations when first mentioned in the text
L 104.”lowered PAR by 67%, 36% and 72% (Figure 2c)” – it should be with positive connotation and explanation that it is change of plant community density
L 105. Fig 2c please change “PAR” by other abbreviation
L 117. Fig 3. “Plant species richness and plant community composition” have no definitions and units in the M&M
L 128-136. Fig. 4. Fungal and bacterial species richness have no definitions and units in the M&M
L 131. Fig. 4- What means OTU?
L 143-144. Again “PAR”
L165. SEM – repeat please transcript here
L 175-186. Fig.6a. Sorry, dear colleagues, but it is not a model: it is something like a graph or network/matrix of correlation coefficients between the parameters in quadrats; (2) the name of your model is “structural EQUATION model” but, as I understood, the system has no core mathematical equations linking the parameters (only a big set of independent regressions in the Supplementary). Moreover it is not clear what mean R2 in all parameters lower N-P-NP on the scheme that are linked not with one but with many other parameters. If it is “the proportion of variations explained” then it should have a specific symbol.
L 193. …aciDIfication…
L 200-204. Bravo! You give the right explanation of the effect of your strange “PAR”! But it shall be not in abiotic factors determining plant growth.
L 222-227. This text shall be moved in M&M in site description.
L 245. I can repeat once more: it is not ecosystem functions, it is e/s parameters (properties). E/s functions are photosynthesis, water and elements consumption, litter, soil OM decomposition and other PROCESSES.
L 269.” The direct and indirect pathways of N and P addition on EMF” – missed “The effect of …“
L. 296. M&M 4.1. Study site and sampling. Please add here a description of vegetation (move it from L 222-227 with addition names of plant communities) AND soil description with taxonomic name by WRB, consequence and thickness of soil horizons; soil texture (sand…loam), pH with organic C and N in A horizon.
L 303-304 Please add more detail for calculation of “PAR” reduction under plant canopy.
L 308-309. The description of NPP determination should be improved by addition of sampling dates: the use of above and belowground biomass as NPP requires sampling at the time of completed growth before the start of biomass decrease. Moreover, usage of root biomass underestimates roots’ NPP due to an active growth but short life span of fine roots.
L 323. M&M has section “4.3. DNA extraction…”. DNA data are absent in the whole Results and Discussion. Please delete this part.
L 327. Section 4.4. Please correct this section as it was discussed above with addition algorithm and functions of EMF calculation.
Supplementary materials, Tables 2-4: the verbal names/symbols of parameters (Plant, also Plants, Bacterial, Fungal, Bacterial composition, Fungal composition, Plant composition, and others) in the tables shall be defined in detail with units (dimentions) and their values (amounts).
The article has comprehensive statistically processed results of new rich experimental data from poorly known region of the Tibetan Plateau. It is a good research in plant ecology. This material shall be published. However the text should be corrected and improved.
1. A clear definition of the ecosystem multifunctionality shall be added in the Introduction with a short explanation on how it can be calculated. Figure 1L has numerical values of EMF, so the section 4.4 in the Material and Methods (M&M) must have functions/equations for EMF calculation.
2. The text contains numerous references to biodiversity, however M&M has no section on how biodiversity was quantitatively expressed, and the results on biodiversity evaluation are not visible in the text. Those seem strange, because this aspect has been under the scrutiny of researchers for a quite long time. Also strange is “belowground biodiversity” that relates only to roots, bacteria and fungi without consideration their taxonomic and functional characteristics. It is a main drawback of the article. Soil fauna as an important component of soil biodiversity is also missed in the article.
3. I was shocked when I saw Figure 2c before reading M&M. I understood from the figure that soil N and P can change solar radiation (PAR,%) going from our Sun! The authors think that they represented data on photosynthetically active radiation (PAR). It is wrong. Two lines in M&M (Section 4.1 L.303-304) show that it is a percentage of light reaching soil surface under plants’ canopy (cover), i.e. it is a measure of shadowing or plants’ density. It is not the abiotic factor; it is a parameter that related to plant community structure. The “PAR” shall be obligatory renamed. The text should be thoroughly corrected with other symbol (f.e. RPAR as relative PAR or RLC relative light conditions). Though, Discussion has a right explanation of the “PAR” effect on plant community (L 200-204).
Some specific comments
The text and supplementary data have many abbreviations for various parameters. I propose giving transcript of the abbreviations in a separate table.
L 18. SEM: please give a transcript of the abbreviation
L 83 “Effects of N and P addition on ecosystem functions”. Actually all numerical data below are parameters (state variables) but not functions (rate variables, flows)
L 86. If no special table for abbreviations, please give transcripts of abbreviations when first mentioned in the text
L 104.”lowered PAR by 67%, 36% and 72% (Figure 2c)” – it should be with positive connotation and explanation that it is change of plant community density
L 105. Fig 2c please change “PAR” by other abbreviation
L 117. Fig 3. “Plant species richness and plant community composition” have no definitions and units in the M&M
L 128-136. Fig. 4. Fungal and bacterial species richness have no definitions and units in the M&M
L 131. Fig. 4- What means OTU?
L 143-144. Again “PAR”
L165. SEM – repeat please transcript here
L 175-186. Fig.6a. Sorry, dear colleagues, but it is not a model: it is something like a graph or network/matrix of correlation coefficients between the parameters in quadrats; (2) the name of your model is “structural EQUATION model” but, as I understood, the system has no core mathematical equations linking the parameters (only a big set of independent regressions in the Supplementary). Moreover it is not clear what mean R2 in all parameters lower N-P-NP on the scheme that are linked not with one but with many other parameters. If it is “the proportion of variations explained” then it should have a specific symbol.
L 193. …aciDIfication…
L 200-204. Bravo! You give the right explanation of the effect of your strange “PAR”! But it shall be not in abiotic factors determining plant growth.
L 222-227. This text shall be moved in M&M in site description.
L 245. I can repeat once more: it is not ecosystem functions, it is e/s parameters (properties). E/s functions are photosynthesis, water and elements consumption, litter, soil OM decomposition and other PROCESSES.
L 269.” The direct and indirect pathways of N and P addition on EMF” – missed “The effect of …“
L. 296. M&M 4.1. Study site and sampling. Please add here a description of vegetation (move it from L 222-227 with addition names of plant communities) AND soil description with taxonomic name by WRB, consequence and thickness of soil horizons; soil texture (sand…loam), pH with organic C and N in A horizon.
L 303-304 Please add more detail for calculation of “PAR” reduction under plant canopy.
L 308-309. The description of NPP determination should be improved by addition of sampling dates: the use of above and belowground biomass as NPP requires sampling at the time of completed growth before the start of biomass decrease. Moreover, usage of root biomass underestimates roots’ NPP due to an active growth but short life span of fine roots.
L 323. M&M has section “4.3. DNA extraction…”. DNA data are absent in the whole Results and Discussion. Please delete this part.
L 327. Section 4.4. Please correct this section as it was discussed above with addition algorithm and functions of EMF calculation.
Supplementary materials, Tables 2-4: the verbal names/symbols of parameters (Plant, also Plants, Bacterial, Fungal, Bacterial composition, Fungal composition, Plant composition, and others) in the tables shall be defined in detail with units (dimentions) and their values (amounts).
Reviewer 3 Report
The manuscript “Above- and belowground biodiversity jointly drives ecosystem multifunctionality after long term field nitrogen and phosphorous addition in alpine meadows on the Tibetan plateau” by Bingheng Cheng et al. analyzes the effect of N and P fertilization, in the context of a long term manipulative experiment, over an alpine meadow discerning the effects of abiotic and biotic changes on ecosystem services and focusing in particular on the relationship between biodiversity and ecosystem multifunctionality. The field of research of the study is of utmost interest in the field of global change ecology. The study concludes that both above and below-ground biodiversity components are more important driving factors in promoting ecosystem multifunctionality than abiotic ones, which is fully supported by the scientific evidence provided. The presented research results are based on solid analytical and statistical methods; furthermore the number of ecological variables considered is very broad, including richness and diversity of plants, fungal, bacterial communities and is key in allowing a comprehensive modelling of the effects of fertilization on alpine grasslands system functionality. The manuscript is well structured, clearly written and provides a very good introduction to the topic and the scientific questions addressed by citing an appropriate and exhaustive list of references.
That said, I strongly recommend the publication of the manuscript on the journal Plants prior to a minor revision which will need to address the following points:
L86: 110% instead of 1.1 times
L91: 120% instead of 1.2 times
L145: richness (add the final “s”)
L200: “N and P addition reduced PAR”. The photosynthetic active radiation (PAR) is (sun)light in the 400-700nm wavelength range which cannot be modified by a fertilization treatment. PAR incoming flux from the sky is unaltered, while the changed grass biomass structure following the fertilization treatment alters the amount of PAR being transmitted through the plants canopy. I suggest rephrasing as: “N and P addition reduced PAR levels in the lower canopy layers”.
L255-257: “However the response of microbial growth to N addition cannot be explained by changes in MBC and microbial growth and respiration were decoupled in soil.” This sentence is not clear. Do the Authors mean that the response microbial growth cannot be ONLY explained by changes in MBC? Since MBC is the results of the contrasting processes of carbon gain in microbial population and C loss through respiration? If so, I suggest rephrasing as: “However the response of microbial growth to N addition cannot be explained only by changes in MBC and microbial growth and soil respiration need to be decoupled.”
L302-305: This sentence is unclear and lacks important methodological information:
1. Sunny day. Do the Authors mean “On sunny days”?
2. Specify exactly the position of sunlight radiation measurements: above the canopy and by the lower layer of the canopy? At which height from the ground were the measurements performed?
3. Provide technical details (Manufacturer, model) of the illumination photometer. Did this radiometer measure PAR fluxes directly? Or did you use a pyranometer to measure global radiation and calculated its PAR component based on a formula? Please provide an explanation of the method used in the text.
Reference #34: (https://ir.lzu.edu.cn/handle/262010/221581). Apparently, there are no files associated to this resource and I could only access the abstract of this study. If the document is not available, consider an alternative appropriate reference.
Round 2
Reviewer 2 Report
Dear Colleagues,
You forgot add "soil" in the title. I propose to change the title as follows: "Species Diversity of Plants and Soil Microorganisms..."
Figure S4: please remove old version of this figure
